# Dynamic Structure-Aware Modulation Network for Underwater Image Super-Resolution

**DOI:** 10.3390/biomimetics9120774

**Published:** 2024-12-19

**Authors:** Li Wang, Ke Li, Chengang Dong, Keyong Shen, Yang Mu

**Affiliations:** 1School of Computer and Software, Nanjing Vocational University of Industry Technology, Nanjing 210023, China; li1019wang@niit.edu.cn (L.W.); dcg18600012570@nuaa.edu.cn (C.D.); 2School of Mechanical and Electrical Engineering, Nanchang Institute of Technology, Nanchang 330044, China; 3School of Mathematics and Statistics, Huangshan University, Huangshan 245021, China; 4School of Computer Information and Engineering, Nanchang Institute of Technology, Nanchang 330044, China; hhu2015hhu@163.com (K.S.); yang_mu_nut@163.com (Y.M.)

**Keywords:** underwater image, super-resolution reconstruction, transformer, spatial structure, feature modulation

## Abstract

Image super-resolution (SR) is a formidable challenge due to the intricacies of the underwater environment such as light absorption, scattering, and color distortion. Plenty of deep learning methods have provided a substantial performance boost for SR. Nevertheless, these methods are not only computationally expensive but also often lack flexibility in adapting to severely degraded image statistics. To counteract these issues, we propose a dynamic structure-aware modulation network (DSMN) for efficient and accurate underwater SR. A Mixed Transformer incorporated a structure-aware Transformer block and multi-head Transformer block, which could comprehensively utilize local structural attributes and global features to enhance the details of underwater image restoration. Then, we devised a dynamic information modulation module (DIMM), which adaptively modulated the output of the Mixed Transformer with appropriate weights based on input statistics to highlight important information. Further, a hybrid-attention fusion module (HAFM) adopted spatial and channel interaction to aggregate more delicate features, facilitating high-quality underwater image reconstruction. Extensive experiments on benchmark datasets revealed that our proposed DSMN surpasses the most renowned SR methods regarding quantitative and qualitative metrics, along with less computational effort.

## 1. Introduction

Due to the refraction, absorption, and scattering of light in underwater scenes, obtained underwater images typically experience significant color distortion, low contrast, and blurred details. Such low-quality images are usually visually poor and may hinder the execution of subsequent tasks of underwater robots, such as target detection, path planning, and ecological monitoring. Accordingly, image SR reconstruction is formulated to restore a high-resolution (HR) image from its corresponding low-resolution (LR) counterpart. Thanks to its capability to efficiently recover high-frequency information, SR technology finds applications in various fields, including remote sensing, medicine, and surveillance.

Of late, the rapid progress in deep learning has given rise to a variety of SR approaches, including convolutional neural network (CNN)-based [1,2,3], generative adversarial network (GAN)-based [4,5,6], and Transformer-based methods [7,8], aimed at enhancing underwater image SR performance. Islam et al. [9] presented a CNN named SRDRM for underwater SR and constructed the SRDRM-GAN using Markovian PatchGAN as a discriminator. Particularly, a large-scale dataset, USR-248, was made available to enhance the training efficacy of both the SRDRM and SRDRM-GAN models. Subsequently, deep simultaneous enhancement and super-resolution (Deep SESR) [10] was presented for learning end-to-end mapping with the UFO-120 dataset. Zhang et al. [11] constructed a multi-path cross-convolution neural network (AMPCNet) that introduced cross-connections among residual blocks and dilated blocks to realize the mutual fusion of diverse features. Similarly, Wang et al. [12] proposed a lightweight multi-stage information distillation network (MSIDN) aimed at optimizing the balance between SR performance and practical applicability. Yang et al. [13] adopted an encoder–decoder model with multiple adaptive feature fusion modules, naming it the lightweight adaptive feature fusion network (LAFFNet) for underwater image enhancement. Sharma et al. [14] devised a wavelength-driven multi-contextual design for deep CNNs for the SR task, termed Deep WaveNet. Although CNN-based methods yield impressive restoration results, the local modeling of CNNs is insensitive to changes in illumination in underwater environments. This limitation hampers the overall improvement of image brightness and contrast.

A recent line of research has delved into Transformer architecture, demonstrating its high effectiveness in underwater tasks by skillfully modeling global dependencies with the use of multiple self-attention mechanisms. Peng et al. [15] reported a U-shape Transformer network for underwater enhancement that strengthens the network’s focus on more attenuated color channels and spatial regions. Mei et al. [16] introduced a simple yet effective network called UIR-Net, a straightforward yet potent architecture crafted to restore crisp visuals from original underwater images plagued by significant particulate impurities and ocean light spots. A novel U-Net-based reinforced Swin-Convs Transformer (URSCT) [17] was presented for simultaneous enhancement and SR, showcasing excellence in capturing global dependencies within nonhomogeneous media distributions. Analogously, a novel Transformer-based block named URTB [18] was devised along with convolutional layers to deal with the color degradation problem. Dharejo et al. [19] presented a novel algorithm called SwinWave-SR, which leverages a Swin Transformer with wavelet blocks to mitigate information degradation during downsampling in a reversible manner. As mentioned above, the Transformer can boost the ability of global feature representation and facilitate the recovery of high-quality images.

The above observations indicate that existing methods can effectively solve the SR problem; however, there are still some issues that deserve to be explored further. On the one hand, the above networks process all degraded images with the same operation, employing a fixed structure, which might lead to sub-optimal results. However, degraded images exhibit different statistics, necessitating the dynamic tuning of a network based on varying input characteristics to achieve optimal results. On the other hand, current hybrid CNN–Transformer methods neglect image structure information, hindering the improvement of image perception quality. Particularly for underwater images with severe degradation, it is crucial to ensure that the model prioritizes the recovery of structural information while capturing rich global features.

In underwater environments, visibility is often hindered by severe light absorption, scattering, and non-uniform degradation, making the human visual system’s ability to dynamically adapt to complex surroundings particularly relevant. Inspired by this, we emulated the multi-scale and adaptive characteristics of human vision to address the challenges of underwater image restoration. The human visual system dynamically processes varying visual stimuli and integrates multi-scale structural information, enabling perception in challenging conditions. This biological insight motivated us to design a model that adapts to diverse input characteristics while capturing both global and structural details critical for underwater scenes.

In this study, we propose a dynamic structure-aware modulation network dubbed DSMN that effectively improves the quality of degraded underwater images. Specifically, the proposed Mixed Transformer incorporates two distinct Transformer blocks to comprehensively explore structure-aware features, enhancing the recovery of underwater image details. Concurrently, we propose a dynamic information modulation module (DIMM) to modulate the output of the Mixed Transformer based on input image statistics, enabling the dynamic adjustment of the network to enhance contrast and color casts effectively. To further address the non-uniform degradation problem, we introduce a hybrid-attention fusion module (HAFM), a unique fusion mechanism designed to efficiently fuse modulated features through multi-dimensional transformation. Comprehensive experiments demonstrated the superiority of our proposed DSMN, outperforming most SR methods while maintaining lower computational complexity.

In short, this study advances three principal contributions.

We propose a lightweight yet effective DSMN for underwater image SR. The network adeptly captures both local and global image features and dynamically adjusts them based on input statistics. Ultimately, it generates high-quality underwater images after undergoing multidimensional processing and fusion.The proposed Mixed Transformer employs a multi-head, structure-aware convolution and multi-head self-attention mechanism to jointly learn local structural attributes and global features, contributing to the better recovery of degraded image details and enhancement of global contrast.The proposed DIMM acquires suitable weights grounded in input statistics, facilitating the dynamic modulation of the Mixed Transformer to accentuate the most information-rich regions.The proposed HAFM explores various feature dimensions and obtains a richer feature content in handling modulation features, resulting in more realistic and accurate underwater images.

## 2. Related Work

### 2.1. Deep Learning for Underwater SR

The emergence of deep learning has led to significant advancements in SR tasks. This progress is primarily attributed to deep learning models trained on extensive datasets, enabling the acquisition of complex feature representations. In the SRDRM [9], the authors employed multiple deep residual multiplier blocks with various convolutional layers, which could recover the global contrast and texture effectively. Deep SESR [10] incorporated dense residual-in-residual sub-networks to facilitate multi-scale hierarchical feature learning, substantially boosting underwater SR performance. The AlphaSR generative adversarial network (AlphaSRGAN) [20] used the AlphaGAN model to learn nonlinear mapping between LR-HR image pairs and yielded better performance in both quantitative and qualitative results. Progressive attentional learning (PAL) [4] adopted a CNN with channel-wise attention to guide the model’s attention toward the most informative regions in the input. In LAFFNet [13], the authors designed an encoder–decoder model that contains multiple adaptive feature fusion modules to acquire semantic information at different scales, as well as the channel attention mechanism to assign weights to three feature maps. Qi et al. [21] presented a novel underwater image enhancement method termed SGUIE-Net, which devises a semantic region-wise enhancement module to learn local enhancement features. Wang et al. [22] constructed an agent-guided, non-local attention network that capitalizes on the integration of the global structure and local context of underwater images, thereby efficaciously augmenting and reinstating detailed visual information. In the work in [23], the authors used the wavelet transform and different convolution kernel sizes to learn image degradation features.

We can observe that the different design techniques of these methods provide varied solutions for the SR task, yielding notably impressive performance. Nevertheless, these methods are constrained to enhancing only local texture details, owing to the local perceptual nature of CNNs. Some SR approaches incorporate GANs to alleviate local modeling deficiencies, but it is worth noting that GAN-based methods often face challenges such as mode collapse and training instability. In contrast, we advocate introducing Transformer structures to model global dependencies and escalate restoration accuracy.

### 2.2. Vision Transformer

As a groundbreaking development, the Transformer model has exhibited remarkable efficacy in the domain of natural language processing, setting a new standard for performance in the field. After its development, the Transformer architecture was adapted for application in the field of computer vision, where it has achieved transformative success. The Vision Transformer (ViT) [24] was the first Transformer method for computer vision, which flattens image patches and passes them into the Transformer. The Swin Transformer, as highlighted in [7], ingeniously constructs a hierarchical feature representation, achieving a significant milestone in computational efficiency. Most recently, a plethora of Transformer-based methods have emerged as state-of-the-art (SOTA) methods in computer vision. A conventional Transformer block is structured with three fundamental elements: multi-head self-attention (MSA), multiple-layer perception (MLP), and layer normalization (LN), in which MSA allows it to focus on the different relationships between image contents and helps to capture long-range dependencies. Leveraging this advantage, Transformer-based methods are progressively integrated into the task of SR. Wang et al. [18] constructed a novel Transformer-based block named URTB to handle the property of color degradation. Peng et al. [15] developed a multi-scale feature fusion Transformer module coupled with a spatial-wise global feature-modeling Transformer module, aimed at enhancing the network’s emphasis on color channels and spatial regions experiencing pronounced attenuation. Similarly, Shen et al. [25] proposed a dual-attention Transformer block that focuses more on severely degraded regions and channels containing more information. In the URSCT [17], the authors seamlessly integrated the Swin Transformer with the U-Net architecture, significantly enhancing its ability to accurately model global dependencies within heterogeneous media distributions.

More research work is currently being conducted on mixing CNNs and Transformers in order to effectively harness both local and global information, thereby significantly elevating the networks’ overall performance. For instance, Huang et al. [26] applied designed adaptive group attention to the Swin Transformer structure, which dynamically catches visual complementary feature channels based on dependencies. Considering the merits of CNNs and Transformers in SR tasks, we devised a Mixed Transformer combined with structural-aware cues to recover high-resolution and high-quality underwater images.

## 3. Proposed Method

### 3.1. Overall Network Architecture

The overall framework of the DSMN is depicted in Figure 1. Given a degraded input image, $X\in R^{H\times W\times 3}$, which is processed by the DSMN, we can acquire an HR image, $DSMN\left(X\right)\in R^{rH\times rW\times 3}$. H×W indicates the spatial dimension and *r* is the scale factor. Firstly, a 3 × 3 convolutional layer is employed to extract the initial features $F^{0}$ and transform the number of channels into *C*. Next, the initial features $F^{0}$ go through two distinct paths. In one path, there is an alternating stack of Mixed Transformers and HAFMs for deep feature extraction. Simultaneously, the other path leads to the DIMM, where dynamic weights based on input attributes are obtained. Normally, the heterogeneous process can be expressed as follows:(1)$$\left\{\begin{array}{c} F^{D}=H_{DFE}\left(F^{0}\right)\\ \xi =H_{DIMM}\left(F^{0}\right) \end{array}\right.$$
where $H_{DFE}\left(\cdot \right)$ is the embedding function, realized alternately by *d* Mixed Transformers, HAFMs, and the modulation operation, as detailed for extracting deep features. $H_{DIMM}\left(\cdot \right)$ is the operation of the DIMM. $F^{D}$ denotes the extracted deeper features and ξ denotes the dynamic weights reflecting the input properties.

Finally, in addressing the underwater SR task, the more complex features $F^{D}$ are scaled up to the desired HR image through the use of a 3 × 3 convolution layer combined with a pixel-shuffle operation. To complement the potential input information, bilinear interpolation is applied to LR inputs, contributing to the sum of HR outputs. The restoration procedure can be characterized as follows:(2)$$DSMN\left(X\right)=H_{UP}\left(F^{D}\right)+H_{Bilinear}\left(X\right)$$
where $DSMN\left(X\right)$ is the final output result. $H_{UP}\left(\cdot \right)$ is the upsample operation and the enhanced operation, implemented by a 3 × 3 convolution and pixel-shuffle operation. $H_{Bilinear}\left(\cdot \right)$ is a bilinear interpolation operation on the input LR image, facilitating an acceleration of the training process.

We trained the network using an $L_{1}$ loss. Given a training dataset, ${\left\{X_{m},Y_{m}\right\}}_{m=1}^{M}$, where *Y* is the target image, the $L_{1}$ loss can be computed as follows:(3)$$L\left(\Theta \right)=\;\frac{1}{M}\sum_{m=1}^{M}{∥Y_{m}-DSMN\left(X_{m}\;\right)∥}_{1}$$
where Θ is the learnable parameter set of the DSMN, and *M* is the number of training images.

### 3.2. Mixed Transformer

A trending topic in computer vision is the advancement of hybrid CNN–Transformer architectures that can complement local and global information well. As known, images often lose detailed information due to light scattering and absorption in underwater environments, while the introduction of local information is instrumental in compensating for these deficiencies. Plus, global information is essential to obtain a full picture of the target object in an image, together with helping to enhance contrast and brightness. Enlightened by this idea, we propose a hybrid architecture dubbed the Mixed Transformer, leveraging both global and local dependencies to enhance the quality of underwater image restoration.

The architecture of the Mixed Transformer is depicted in Figure 2b, which is composed of two consecutive Transformer blocks, named the structure-aware Transformer block and multi-head Transformer block. It can be discerned that the essence of the structure-aware Transformer block is encapsulated within multi-head structure convolution (MSC), whereas the foundational aspect of the multi-head Transformer block is rooted in multi-head self-attention (MSA). We can express the output, $F_{Mix}^{d}$, of the *d*-th Mixed Transformer as follows:(4)$${\bar{F}}_{MSC}^{d}=MSC\left(LN\left(F^{d}\right)\right)+F^{d}$$
(5)$$F_{MSC}^{d}=MLP\left(LN\left({\bar{F}}_{MSC}^{d}\right)\right)+{\bar{F}}_{MSC}^{d}$$
(6)$${\bar{F}}_{MSA}^{d}=MSA\left(LN\left(F_{MSC}^{d}\right)\right)+F_{MSC}^{d}$$
(7)$$F_{Mix}^{d}=MLP\left(LN\left({\bar{F}}_{MSA}^{d}\right)\right)+{\bar{F}}_{MSA}^{d}$$
where ${\bar{F}}_{MSC}^{d}$, $F_{MSC}^{d}$, ${\bar{F}}_{MSA}^{d}$, and ${\bar{F}}_{MSA}^{d}$ are the outputs of MSC (MSA) and MLP. MSA is the core of the standard Transformer that catches long-range dependencies effectively.

The multi-head Transformer block (Figure 2a) captures global features more effectively, allowing the model to process a broad range of contextual information. However, in heavily degraded underwater images, an overemphasis on global features may result in the loss of local details. To compensate for this shortcoming, we propose a structure-aware Transformer block based on MSC, which incorporates multiple asymmetric convolutions with different kernel sizes. This design allows MSC to capture diverse spatial structural features across multiple scales while concurrently reducing the computational load. More specifically, multiple asymmetric convolutions can investigate multi-granularity information in both the horizontal and vertical directions of an image, thereby enhancing the network’s sensitivity to spatial structure. Figure 2b provides a detailed view of MSC, which divides input channels into *N* heads and then applies a distinct asymmetric convolution to each head.

Let the Mixed Transformer input features be $F^{d}\in^{\;H\times W\times C}$, which pass through linear layers to generate the values *M* and *V*. On the one hand, the value *V* applies a 3 × 3 depth-wise separable convolution to improve the feature representation further, referred to as V. On the other hand, we split the value *M* into multiple heads along the channel dimension, denoted by $f=\left\{f_{1},f_{2},\ldots ,f_{N}\right\}\in^{\;H\times W\times \frac{C}{N}}$. After that, each head undergoes processing by its unique asymmetric convolution, enabling adaptive focus on spatial structural features at different granularities. Finally, all spatial structural features are processed via a concatenation operation, followed by two successive 1 × 1 convolutions to attain an output feature map, M. For the sake of brevity, we omit the description of the activation operation here. The proposed MSC can be obtained as follows:(8)$$MSC\left(F_{d}\right)=f_{1\times 1}^{1}\left(f_{1\times 1}^{2}\left(\left[C_{k_{1}}\left(f_{1}\right),\cdots ,C_{k_{N}}\left(f_{N}\right)\right]\right)\right)$$
where $f_{1\times 1}\left(\cdot \right)$ denotes a 1 × 1 convolution operation. $C_{k_{n}}\left(\cdot \right)$ is the embedding function, performed by $k_{n}\times 1$ and $1\times k_{n}$ convolutions, in which the kernel size $k_{n}\in \left\{3,5,\ldots ,K\right\}$ increases monotonically by 2 per head.

As illustrated in Figure 2b, we adopted the output feature map M to modulate the value V, utilizing a scalar product. As a result, we could acquire the final output *Z* of MSC as follows:(9)Z=M⊙V
where ⊙ means the element-wise multiplication.

### 3.3. Dynamic Information Modulation Module (DIMM)

To further strengthen the accuracy of image restoration, we propose that the DIMM obtains appropriate weights based on input statistics, achieving the dynamic modulation of the Mixed Transformer. More importantly, the acquisition of correlation information through learning assists the network in gaining a better understanding of the distribution of light and color, thereby facilitating the improvement of brightness and color rendition in underwater imagery. Figure 1 illustrates the structure and output signal flow of the DIMM, comprising two 3 × 3 convolutions, a ReLU, and a global pooling operation. Formally, the DIMM determines the Mixed Transformer weight parameters as follows:(10)ξ=HDIMMF0=glof3×31ReLUf3×32F0=ξ1,⋯,ξd
where $glo\left(\cdot \right)$ is the global average pooling operation. $f_{3\times 3}\left(\cdot \right)$ denotes a 3 × 3 convolution operation. The dynamic modulation process of the *d*-th Mixed Transformer can be formulated as:(11)$$F_{Mou}^{d}=\xi^{d}\cdot F_{Mix}^{d}$$
Throughout the training process, the DIMM learns the correlation between input image statistics and the Mixed Transformer. This learning mechanism helps the network to utilize appropriate features from various combinations of primitive and advanced structural features.

### 3.4. Hybrid-Attention Fusion Module (HAFM)

Acknowledging the significant impact of heterogeneous degradation in underwater scenes on image restoration quality underscores the necessity for more intricate content representations to achieve enhanced performance. Accordingly, we propose the HAFM to capture discriminative features in multiple dimensions, improving feature representativeness while extracting abundant spatial information. As depicted in Figure 3, the HAFM first concentrates on two branches, spatial and channel information, and subsequently performs multiplication to dynamically adapt to different structures and contents in the image. Finally, a Softmax operation is utilized to generate distinct weight distributions, adaptively adjusting the enhanced features produced by the third branch. This approach enables the model to synthesize information from different dimensions, facilitating the handling of the diversity and complexity in underwater images. Specifically, for the input modulation features $F_{Mou}^{d}\in^{\;H\times W\times C}$, the first branch output features $F_{sp}^{d}\in^{\;H\times W\times 1}$, second branch output features $F_{ch}^{d}\in^{\;1\times 1\times C}$, and third branch output features $F_{sc}^{d}\in^{\;H\times W\times C}$ can be expressed as follows:(12)$$F_{sp}^{d}=H_{emd}\left(F_{Mou}^{d}\right)$$
(13)$$F_{ch}^{d}=H_{emd}\left(avg\left(F_{Mou}^{d}\right)\right)$$
(14)$$F_{sc}^{d}=H_{emd}\left(F_{Mou}^{d}\right)$$
where $H_{emd}\left(\cdot \right)$ is the embedding function, performed by a 3 × 3 convolution, ReLU, and a 1 × 1 convolution. $avg\left(\cdot \right)$ denotes an average pooling operation. After that, a multiplication operation is performed on these features, which can be defined as follows:(15)$$F^{d}=Softmax(F_{sp}^{d}⊗F_{ch}^{d})⊗F_{sc}^{d}=\varpi^{d}⊗F_{sc}^{d}$$
where ⊗ is Kronecker product and $Softmax\left(\cdot \right)$ is the Softmax function. $\varpi^{d}$ indicates the weight coefficients that are employed to recalibrate $F_{sc}^{d}$.

## 4. Experiments

### 4.1. Dataset and Experimental Setup

Our study applied the proposed methodology to publicly accessible underwater image datasets, namely USR-248 [9] and UFO-120 [10]. The USR-248 dataset comprised 1060 pairs of training samples and 248 pairs of testing samples. HR images were subjected to downsampling using bicubic interpolation at scale factors of ×2, ×4, and ×8, coupled with the introduction of 20% Gaussian noise. The HR image had a size of 640 × 480, with corresponding LR sizes of 320 × 240, 160 × 120, and 80 × 60. The UFO-120 dataset comprised 1500 pairs of training samples and 120 pairs of testing samples, employing scale factors of ×2, ×3, and ×4. The HR image had a size of 640 × 480, with corresponding distorted LR sizes of 320 × 240, 213 × 160, and 160 × 120. Figure 4a and Figure 4b depict a particular instance with the USR-248 and UFO-120 datasets, respectively. To be consistent with mainstream algorithms, the peak signal-to-noise ratio (PSNR), structure similarity index (SSIM), and underwater image quality measure (UIQM) [27] were evaluated for the proposed network.

An Adam optimizer was employed to minimize the objective function, configuring the optimizer parameters as follows: $\beta_{1}=0.9$; $\beta_{2}=0.999$; and $\varepsilon =10^{-8}$. The learning rate was initially set to 1 × 10^−3^ and was halved every 200 epochs. Each batch consisted of 32 LR patches of the size 50 × 50 for the SR task. To build a lightweight model, we set the number of channels (*C*) to 64, the number of heads (*N*) to 4, and the kernel size (*K*) to 9. Table 1 presents the hyperparameters utilized in our network architecture. We performed our method employing the PyTorch framework with NVIDIA RTX 3090 GPU.

### 4.2. Ablation Study

To illustrate how the proposed components enhanced the model’s performance, we conducted ablation experiments with the USR-248 dataset. We sequentially removed the DIMM, Mixed Transformer, and HAFM for retraining. Notably, to accommodate the DIMM modulation, we replaced the structure-aware Transformer block with a multi-head Transformer block, given the specificity of the Mixed Transformer. The ablation results can be seen in Table 2. Bold is used to highlight the best performance metrics, whereas the FLOPs was measured at 640 × 480 per HR image.

**(1) Effect of DIMM.** The DIMM obtained different weights based on the input information and then realized the dynamic modulation of the Mixed Transformer, thus enhancing the network’s representational capability. The model without the DIMM support exhibited a performance degradation of 0.13, 0.08, and 0.06 dB compared to the DSMN, respectively. Moreover, we provide visualizations of average feature maps for different models in Figure 5. For the color intensity, red indicates a large value while blue indicates a small one. It is evident that the information provided by the DSMN without the DIMM was less informative and was not conducive to generating high-frequency details.

**(2) Effect of Mixed Transformer.** The Mixed Transformer explored global information to strengthen image contrast and brightness, where the structure-aware Transformer block could efficiently learn spatial structure cues. As evident from Table 2, the difference between the DSMN and DSMN without the Mixed Transformer was not significant in terms of objective indicators. Furthermore, we employed the Canny [28] algorithm to assess the improvement in image sharpness in Figure 6. Abundant edge information facilitated object recognition and segmentation, enhancing the understanding of the overall scene. Leveraging the Mixed Transformer, the DSMN could detect additional structural textures along the edges.

**(3) Effect of HAFM.** The HAFM could capture multi-dimensional discriminative features that could extract abundant spatial information. The DSMN augmented by the HAFM exhibited an increase in quantitative metrics by 0.73%, 18%, and 3.73%, respectively. Figure 5 and Figure 6 indicate the advantages of the HAFM in spatial information capture.

**(4) Numbers of Mixed Transformers.** Herein, we illustrate the effect of the network depth (change in *d* = 1, 2, 3, 4, and 5) on the PSNR and SSIM in Table 3 and Figure 7. The model performance improved with an increase in the depth, but the improvement slowed down as *d* exceeded 4. Recognizing the trade-off between model operations and accuracy, we opted for *d* = 4 as the optimal network depth.

### 4.3. Comparison to SOTA Methods

#### 4.3.1. Results with the USR-248 Dataset

We benchmarked our model against several cutting-edge methodologies using the USR-248 dataset, containing the SRCNN [29], VDSR [30], EDSRGAN [31], SRGAN [32], SRResNe [32], ESRGAN [33], SRDRM [9], SRDRM-GAN [9], PAL [4], AMPCNet [11], Deep waveNet [14], and RDLN [34] algorithms. Among them, the first six algorithms are designed for natural image SR tasks, while the last six algorithms are tailored for underwater image restoration. We evaluated the methodology using both quantitative and qualitative metrics, where the quantitative metric provided objective evaluations and the qualitative metric assessed visual quality.

Table 4 systematically lists the experimental results of our investigation. An insightful examination of these results reveals that our proposed DSMN acquired the highest PSNR score across all scales. The DSMN demonstrated improvements of at least 0.70%, 0.57%, and 1.63% with the scale factors ×2, ×4, and ×8, respectively. In the case of the SSIM, our DSMN continued to achieve the best results with the scale factors ×2 and ×4. Although the proposed method did not obtain optimal results with the UIQM, it remained competitive when compared to recently proposed works such as the Deep WaveNet and RDLN. This is because other methods utilized a larger number of learnable parameters to achieve superior performance, whereas our method, with less than 0.7 M parameters, delivered competitive quantitative results. Furthermore, we provide a visual comparison of different algorithms in Figure 8. It is evident that the images recovered by our proposed DSMN are more realistic and natural. For instance, at the scale factors ×2 and ×4, the visual effect of PAL is similar to that of our method, closely resembling the real image in terms of details and contrast. However, at the scale factor ×8, the PAL-recovered image not only appears distorted but also exhibits oversaturation. In contrast, the proposed DSMN accurately reconstructs stripe details and enhances the global contrast of the image, showcasing the network’s robust capabilities in capturing both global and spatial structures.

Figure 9 displays an intuitive comparison of the model capacity and results with USR-248 dataset, with our method labeled by a red star. Our proposed DSMN yielded the highest PSNR and SSIM scores with a lower number of parameters and FLOPs. Whilst the UIQM was slightly lower than the SRDRM and SRDRM-GAN, our method achieved reductions in the number of parameters by 33.7% and 95.1%, respectively.

#### 4.3.2. Results with the UFO-120 Dataset

To further exhibit the effectiveness of our DSMN, we compared it to some well-known models with the UFO-120 dataset. As shown in Table 5, all methods except the SRCNN and SRGAN are specifically tailored for underwater image restoration.

Quantitative and qualitative results are, respectively, displayed in Table 5 and Figure 10. Compared to the mainstream models, our DSMN yielded superior results at different scales. In the PSNR, the proposed method was 0.09 dB lower than the SRGAN but offered the highest scores at the scale factors ×3 and ×4. In terms of the SSIM and UIQM, our DSMN procured supremacy at the scale factors ×2 and ×3, while Deep WaveNet outperformed our method by 0.02 and 0.08 at the scale factor ×4. As depicted in Figure 10, Deep WaveNet produced an obvious mosaic effect, and AMPCNet failed to recover details and correct color bias. In contrast, our model possessed better visual effects, reconstructing rich details, correcting color bias, and enhancing contrast.

To better illustrate the efficacy and superiority of the proposed approach, we employed a local attribution map (LAM) [35] with an integral gradient method to explore which input pixels contributed most to the final performance. As depicted in Figure 11, the red marked points represent information pixels that aided in image restoration, with higher diffusion index (DI) values indicating a broader range of attention. It can be seen that the DSMN produced the most focused results, with the clearest information on the focal areas and the highest DI values, showing the best ability to reconstruct the original details.

## 5. Conclusions

In this study, an effective SR method, dubbed the DSMN, was proposed for restoring high-resolution and high-quality underwater images. Firstly, we proposed a Mixed Transformer, incorporating a structure-aware block and multi-head block, to abstract richer local and global features for global contrast and luminance enhancement. Then, aiming at better adapting to the input data characteristics, we designed a DIMM that modulated the Mixed Transformer by generating different weights to emphasize information-rich regions. Finally, the HAFM was devised to catch more delicate features using spatial and channel interaction for high-quality image reconstruction. Extensive experiments indicated that our method achieved competitive performance compared to mainstream methods, and ablation studies further demonstrated the effectiveness of our proposed components. Despite the strong performance of our DSMN, it has certain limitations. For instance, the model’s reconstruction, particularly in terms of structure and detail, requires further improvement after recovery using our method. Moreover, the fixed up-sampling strategy fails to adaptively adjust the magnification process based on the image content. Future work will focus on addressing these limitations.

## Figures and Tables

**Figure 1 biomimetics-09-00774-f001:**
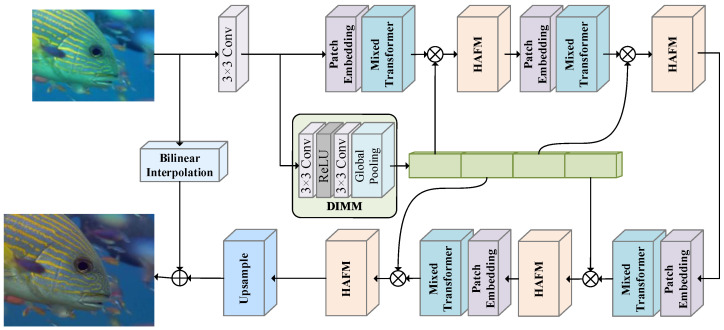
The architecture of our proposed DSMN, which consists of a DIMM, Mixed Transformers, and HAFMs to progressively gather features rich in detail and enhance contrast.

**Figure 2 biomimetics-09-00774-f002:**
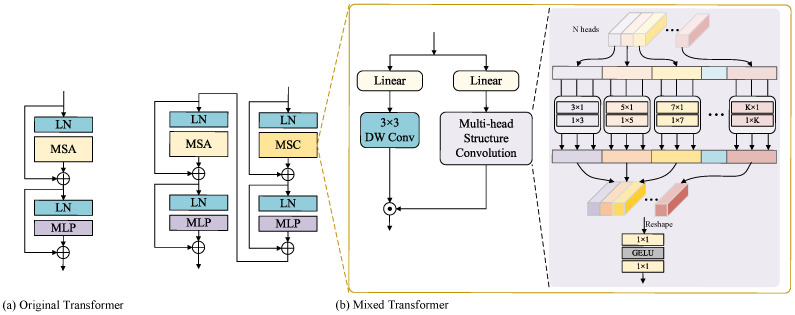
(**a**) Original Transformer with MSA. (**b**) Mixed Transformer that contains structure-aware Transformer block and multi-head Transformer block. MSC that aggregates multiple asymmetric convolutions with different kernel sizes.

**Figure 3 biomimetics-09-00774-f003:**
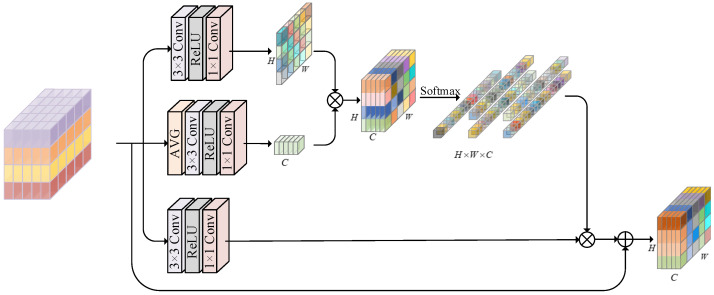
The architecture of the HAFM that autonomously aggregates discriminative features in multiple dimensions.

**Figure 4 biomimetics-09-00774-f004:**
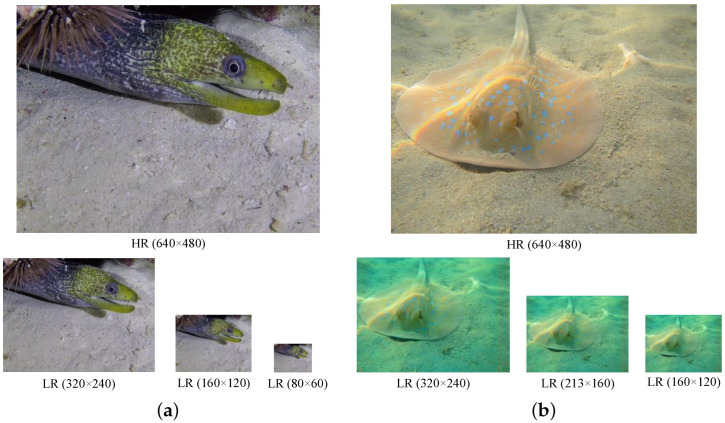
Particular instances with the USR-248 and UFO-120 datasets. (**a**) The USR-248 dataset facilitates paired training with scale factors of ×2, ×4, and ×8. (**b**) The USR-248 dataset facilitates paired training with scale factors of ×2, ×3, and ×4.

**Figure 5 biomimetics-09-00774-f005:**
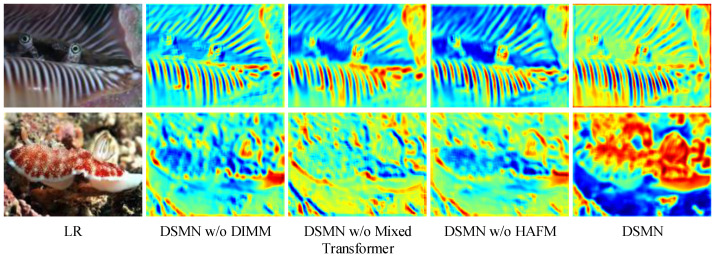
Visualized feature maps of different models. It is evident that, enhanced by the DIMM, Mixed Transformer, and HAFM, our DSMN demonstrated a stronger response in the target area, highlighting its robust information capture capabilities.

**Figure 6 biomimetics-09-00774-f006:**
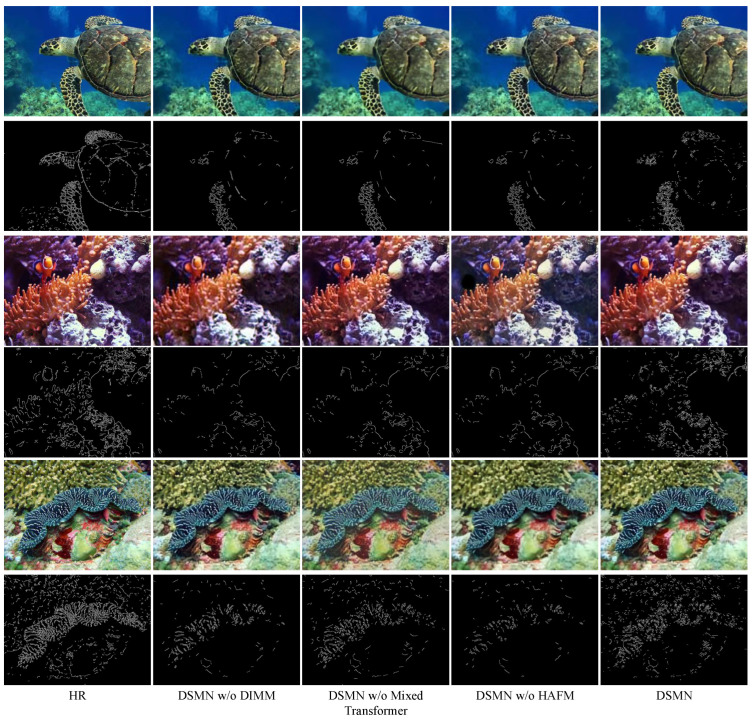
Canny edge detection of different models. DSMN is capable of detecting detailed edge and structural texture information.

**Figure 7 biomimetics-09-00774-f007:**
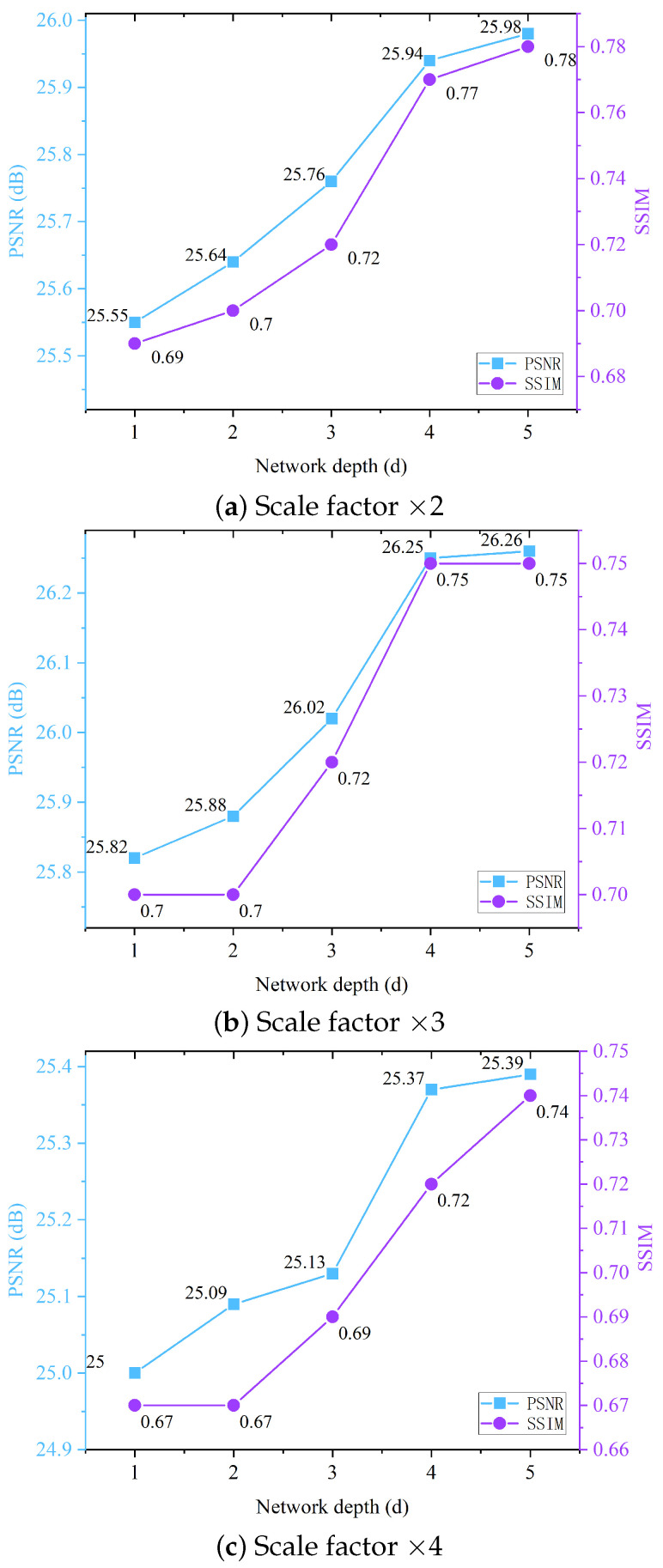
PSNR/SSIM results achieved with UFO-120 dataset as network depth (*d*) increased. When *d* exceeded 4, the PSNR and SSIM performance gains diminished.

**Figure 8 biomimetics-09-00774-f008:**
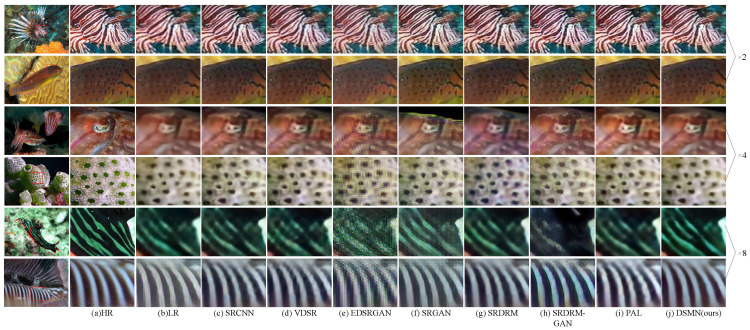
Visual comparison of our proposed DSMN against popular works with USR-248 dataset.

**Figure 9 biomimetics-09-00774-f009:**
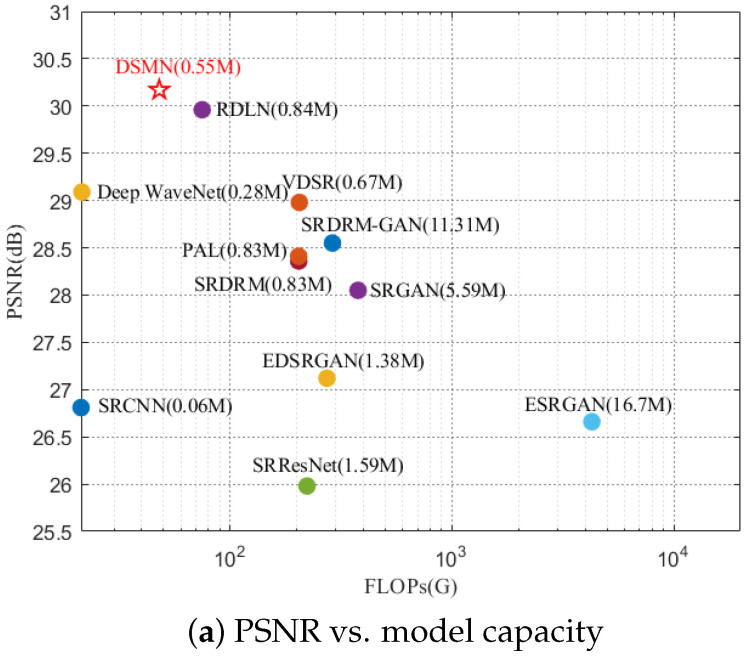
Comparison of model capacity and performance between our DSMN (red star) and dominant methods with USR-248 for scale factor ×2. One can see that our DSMN effectively balanced high accuracy and model capacity.

**Figure 10 biomimetics-09-00774-f010:**
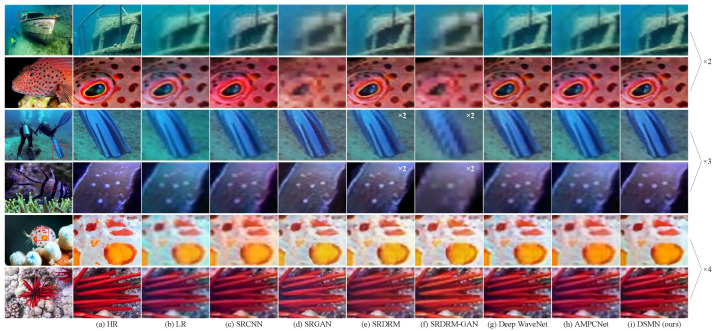
Visual comparison of our proposed DSMN against popular works with UFO-120 dataset.

**Figure 11 biomimetics-09-00774-f011:**
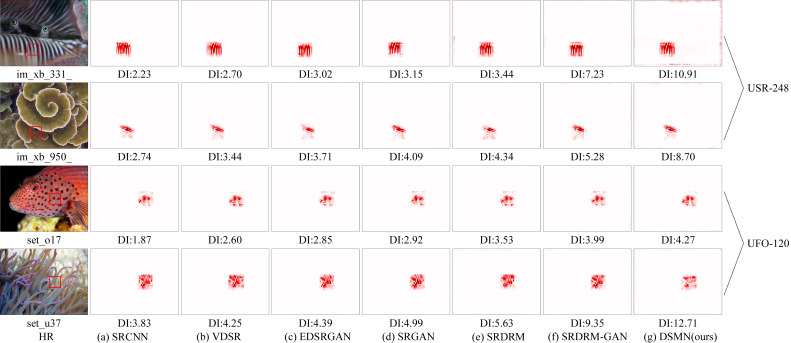
Comparison of LAM attribution results against popular works for scale factor ×4. It can be seen that our DSMN pixels are informative and obtain the highest DI value.

**Table 1 biomimetics-09-00774-t001:** Values of hyperparameters for our method.

Hyperparameters	Value
Use_hflip	True
Use_rot	True
Use_shuffle	True
Optim_g	Adam
Number of channels (*C*)	64
Number of heads (*N*)	4
Kernel size (*K*)	9
LR	0.001
Weight_decay	0
Betas	[0.9, 0.999]
Gamma	0.5
Loss type	L1

**Table 2 biomimetics-09-00774-t002:** Ablation study of different components with USR-248 for scale factor ×4. Higher PSNR, SSIM, and UIQM values reflect higher image quality.

Model	FLOPs (G)	Params (M)	PSNR (dB)	SSIM	UIQM
DSMN w/o DIMM	11.6	0.53	26.15	0.57	2.44
DSMN w/o Mixed Transformer	10.7	0.47	26.24	0.62	2.47
DSMN w/o HAFM	10.9	0.45	26.09	0.55	2.41
DSMN	12.4	0.57	**26.28**	**0.65**	**2.50**

**Table 3 biomimetics-09-00774-t003:** Average PSNR (dB)/SSIM with UFO-120 dataset for different network depths (*d*).

Network Depth *d*	FLOPs (G)	Params (M)	×2	×3	×4
1	12.32	0.23	25.55/0.69	25.82/0.70	25.00/0.67
2	23.79	0.33	25.64/0.70	25.88/0.70	25.09/0.67
3	36.86	0.42	25.76/0.72	26.02/0.72	25.13/0.69
4	47.98	0.55	25.94/0.77	26.25/0.75	25.37/0.72
5	59.01	0.68	25.98/0.78	26.26/0.75	25.39/0.74

**Table 4 biomimetics-09-00774-t004:** Quantitative results with the USR-248 dataset.

Scale	Method	FLOPs (G)	Params (M)	PSNR (dB)	SSIM	UIQM
×2	SRCNN [29]	21.3	0.06	26.81	0.76	2.74
	VDSR [30]	205.28	0.67	28.98	0.79	2.57
	EDSRGAN [31]	273.34	1.38	27.12	0.77	2.67
	SRGAN [32]	377.76	5.95	28.05	0.78	2.74
	SRResNet [32]	222.37	1.59	25.98	0.72	-
	ESRGAN [33]	4274.68	16.7	26.66	0.75	2.70
	SRDRM [9]	203.91	0.83	28.36	0.80	**2.78**
	SRDRM-GAN [9]	289.38	11.31	28.55	0.81	2.77
	PAL [4]	203.82	0.83	28.41	0.80	-
	AMPCNet [11]	-	1.15	29.54	0.80	2.77
	Deep WaveNet [14]	21.47	0.28	29.09	0.80	2.73
	RDLN [34]	74.86	0.84	29.96	0.83	2.68
	DSMN (ours)	47.98	0.55	**30.17**	**0.84**	2.75
×4	SRCNN [29]	21.3	0.06	23.38	0.67	2.38
	VDSR [30]	205.28	0.67	25.70	0.68	2.44
	EDSRGAN [31]	206.42	1.97	21.65	0.65	2.40
	SRGAN [32]	529.86	5.95	24.76	0.69	2.42
	SRResNet [32]	85.49	1.59	24.15	0.66	-
	ESRGAN [33]	1504.09	16.7	23.79	0.66	2.38
	SRDRM [9]	291.73	1.90	24.64	0.68	2.46
	SRDRM-GAN [9]	377.2	12.38	24.62	0.69	2.48
	PAL [4]	303.42	1.92	24.89	0.69	-
	AMPCNet [11]	-	1.17	25.90	0.66	**2.58**
	Deep WaveNet [14]	5.59	0.29	25.20	0.68	2.54
	RDLN [34]	29.56	0.84	26.16	0.66	2.38
	DSMN (ours)	12.39	0.57	**26.31**	**0.70**	2.53
×8	SRCNN [29]	21.3	0.06	19.97	0.57	2.01
	VDSR [30]	205.28	0.67	23.58	0.63	2.17
	EDSRGAN [31]	189.69	2.56	19.87	0.58	2.12
	SRGAN [32]	567.88	5.95	20.14	0.60	2.10
	SRResNet [32]	51.28	1.59	19.26	0.55	-
	ESRGAN [33]	811.44	16.7	19.75	0.58	2.05
	SRDRM [9]	313.68	2.97	21.20	0.60	2.18
	SRDRM-GAN [9]	399.15	13.45	20.25	0.61	2.17
	PAL [4]	325.51	2.99	22.51	**0.63**	-
	AMPCNet [11]	-	1.25	23.83	0.62	**2.25**
	Deep WaveNet [14]	1.62	0.34	23.25	0.62	2.21
	RDLN [34]	18.23	0.84	23.91	0.54	2.18
	DSMN (ours)	3.50	0.65	**24.30**	0.55	2.19

**Table 5 biomimetics-09-00774-t005:** Quantitative results with the UFO-120 dataset.

Method	PSNR (dB)			SSIM			UIQM		
	×2	×3	×4	×2	×3	×4	×2	×3	×4
SRCNN [29]	24.75	22.22	19.05	0.72	0.65	0.56	2.39	2.24	2.02
SRGAN [32]	**26.11**	23.87	21.08	0.75	0.70	0.58	2.44	2.39	2.56
SRDRM [9]	24.62	-	23.15	0.72	-	0.67	2.59	-	2.57
SRDRM-GAN [9]	24.61	-	23.26	0.72	-	0.67	2.59	-	2.55
AMPCNet [11]	25.24	25.73	24.70	0.71	0.70	0.70	2.93	2.85	2.88
Deep WaveNet [14]	25.71	25.23	25.08	0.77	**0.76**	**0.74**	2.99	2.96	**2.97**
URSCT [17]	25.96	-	23.59	**0.80**	-	0.66	-	-	-
RDLN [34]	25.96	**26.55**	25.37	0.76	0.74	0.73	2.98	**2.98**	2.94
DSMN (ours)	26.02	26.42	**25.59**	**0.80**	**0.76**	0.72	**3.00**	**2.98**	2.89

## Data Availability

The original contributions presented in the study are included in the article, further inquiries can be directed to the corresponding authors.

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
