# Peer review of "Dynamic Structure-Aware Modulation Network for Underwater Image Super-Resolution"

_biomimetics, 2024, doi:10.3390/biomimetics9120774_

Round 1
Reviewer 1 Report
Comments and Suggestions for Authors
The introduction successfully sets the scene and gives enough background information. Nonetheless, think about quickly highlighting the uniqueness of the suggested DSMN in contrast to current state-of-the-art techniques.
Although the methodology is sound, reproducibility might be improved with more information on dataset pretreatment, hyperparameter adjustment, and training computational resources.
It would be beneficial to include a discussion of the approach's possible drawbacks, such as its applicability to non-underwater circumstances or other picture restoration tasks.
Clear and well-presented, the results provide thorough comparisons. But adding statistical analysis (such significance testing) could make the arguments for DSMN's superiority stronger.
Annotations emphasizing significant advancements over baseline techniques in the figures would enhance the effectiveness of the visualizations.
The citations are pertinent and appropriate. To further confirm the work's present relevance, think about referencing more recent research, if any.
For clarity, make sure each figure is properly labeled and scaled. The accessibility of the study might be improved by including a summary table that compiles the most important findings from several datasets.
Reviewer 2 Report
Comments and Suggestions for Authors
The Title: “Dynamic Structure-aware Modulation Network for Underwater Image Super-Resolution”
This work proposes a dynamic structure-aware modulation network (DSMN) for efficient and accurate underwater SR. Mixed Transformer incorporates structure-aware Transformer block and multi-head Transformer block, which can comprehensively utilize local structural attributes and global features to enhance the details of underwater image restoration. The work is interesting in its field, and the authors have faithfully reflected on their work. However, there are some points that must be addressed which are as follows:
1-The related work section should include some recent works from 2024. In addition, the included works need further discussion and analysis on their methodologies and findings.
2- Please cite any specific figures, dataset, equation, and information with reliable sources unless they are related to the authors.
3- The proposed work used two consecutive Transformer blocks, which are the structure-aware transformer block and the multi-head transformer block. It is recommended to justify the specific reasons for their use and give more details about their structure.
4- Define the criteria that were used to determine the improved performance? For example, the criteria used in Table 1 for evaluation purposes should be specified to demonstrate which performance criteria have improved of the proposed work.
5- Why two metrics, quantitative and qualitative are used for the evaluation purpose? Please give more clarification of this point.
Besides, based on UFO-120 dataset, it is stated that “our model possesses better visual effects, reconstructing rich details, correcting color bias, and enhancing contrast”. How are each of these criteria has been verified?
6- The conclusion section needs further discussion on the results obtained and future directions should be added to it.
Reviewer 3 Report
Comments and Suggestions for Authors
This paper proposes a method for underwater image super-resolution based on several modules that combine convolutional and transformer layers. Performance is evaluated to be similar or better to several other methods that were also developed to deal with underwater images.
The following points would increase the clarity of the paper and its results.
- In equation 2, the variable S does not seem to be defined.
- From equation 3, the number of training images seems to be N, but on line 180, it is set to m.
- Elementwise multiplication is defined as using both the circle-dot (line 225) and circle-cross (line 261) notation; the latter usually refers to the tensor product or Kronecker product.
- While PSNR and SSIM are relatively common metrics, UIQM is a more uncommon metric that is defined in a closed-access journal, so it should be at least referenced, and adding additional information, such as the range of the metric, would make the results easier to understand.
- Information in the tables, such as the placement of units for the number of parameters and FLOPs, have inconsistent notation.
- The meaning of the colormap in figure 4 is unclear.
- When possible, both LR and SR figures should be visible in the visual comparisons between methods for easier comparison.
- The meaning of the inclusion of edge detection in figure 5 is unclear.
- An inclusion of a wider selection of network depths in figure 6, as well as starting the graphs from zero, would make for a clearer understanding of the network dynamics.
- Rather than comparing models by FLOP performance in figure 8, comparision with respect to model size would be more consistent with the discussion in the paper.
- The model selection criteria for comparison in figures 7 and 9 are unclear.
Comments on the Quality of English LanguageSome light proofreading would improve the quality of this paper. Most notable is the following:
- a lack of spacing between punctuation and adjacent text
- occasional sentence fragments
- long paragraphs that can be broken up into shorter paragraphs on a single topic (for example, 2.2)
- misspelling or inconsistent capitalization of several important terms, such as "softmax," "convolutional," and "mixed transformer"
